# Informing prevention of stillbirth and preterm birth in Malawi: development of a minimum dataset for health facilities participating in the DIPLOMATIC collaboration

Beatrix von Wissmann [1,2] Elizabeth Wastnedge,[3] Donald Waters [3]
Luis A Gadama,[4] Queen Dube,[5] Clemens Masesa,[6] Elizabeth Chodzaza,[7]
Sarah Jane Stock [8] Rebecca M Reynolds,[9] John Norrie,[10] Enita Makwakwa,[11]
Bridget Freyne,[5,12] Harry Campbell,[13] Jane E Norman [14] Rachael Wood [1,15]
On behalf of the DIPLOMATIC collaboration

For numbered affiliations see end of article.

**Correspondence to**
Dr Beatrix von Wissmann;
beatrix.vonwissmann@ggc.scot. nhs.uk

## ABSTRACT

**Objective** The global research group, DIPLOMATIC (Using eviDence, Implementation science, and a clinical trial PLatform to Optimise MATernal and newborn health in low Income Countries), aims to reduce stillbirths and preterm births and optimise outcomes for babies born preterm. Minimum datasets for routine data collection in healthcare facilities participating in DIPLOMATIC (initially in Malawi) were designed to assist understanding of baseline maternal and neonatal care processes and outcomes, and facilitate evaluation of improvement interventions and pragmatic clinical trials.

**Design** Published and grey literature was reviewed alongside extensive in-country consultation to define relevant clinical best practice guidance, and the existing local data and reporting infrastructure, to identify requirements for the minimum datasets. Data elements were subjected to iterative rounds of consultation with topic experts in Malawi and Scotland, the relevant Malawian professional bodies and the Ministry of Health in Malawi to ensure relevance, validity and feasibility.

**Setting** Antenatal, maternity and specialist neonatal care in Malawi.

**Results** The resulting three minimum datasets cover the maternal and neonatal healthcare journey for antenatal, maternity and specialist neonatal care, with provision for effective linkage of records for mother/baby pairs. They can facilitate consistent, precise recording of relevant outcomes (stillbirths, preterm births, neonatal deaths), risk factors and key care processes.

**Conclusions** Poor quality routine data on care processes and outcomes constrain healthcare system improvement. The datasets developed for implementation in DIPLOMATIC partner facilities reflect, and hence support delivery of, internationally agreed best practice for maternal and newborn care in low-income settings. Informed by extensive consultation, they are designed to integrate with existing local data infrastructure and reporting as well as meeting research data needs. This work provides a transferable example of strengthening data infrastructure

## Strengths and limitations of this study

► In line with recommended practice, the datasets were designed using a multimodal approach, combining an in-depth review of the contextual information and published and grey literature with expert input.

► Delphi methods to gain expert consensus were considered, but deemed unsuitable due to the large number of variables, many of which form interconnected constructs and could thus not be changed independently of each other.

► Instead of a Delphi survey, iterative rounds of consultation with project collaborators and other stakeholders in Malawi were undertaken, which were essential to facilitate Malawian ownership, increasing the likelihood that study recommendations can be implemented and scaled up to increase impact.

► One limitation of this inclusive approach is that the datasets are more extensive than other exemplars of national minimum datasets.

► However, the approach ensures that the datasets can integrate with existing systems and meet local data requirements, and also have the capacity to evaluate other Ministry of Health initiatives, rather than addressing research requirements only.

to underpin a learning healthcare system approach in low-income settings.

DIPLOMATIC is funded by the UK National Institute for Health Research.

## INTRODUCTION

The global burden of stillbirths and preterm births remains significant. According to recent global estimates, 2.6 million babies were stillborn at 28 completed weeks' gestation or over in 2015.[1] Half of stillborn babies



died during labour.[1] Complications of preterm birth (defined as delivery before 37 completed weeks' gestation) accounted for nearly one-fifth of deaths in under 5-year-old children globally (18%) in 2016.[2] Babies born preterm, who survive, remain at risk of developmental delay[3] and long-term physical morbidities.[4]

## DIPLOMATIC

DIPLOMATIC (Using eviDence, Implementation science, and a clinical trial PLatform to Optimise MATernal and newborn health in low Income Countries) is a National Institute of Health Research-funded global research group with partners in Malawi, Zambia and the UK, which aims to reduce stillbirth and preterm birth, and optimise outcomes for babies born preterm. In Malawi and Zambia, an estimated 14 000 and 13 000 stillbirths occur every year (21.8 and 20.9 stillbirths per 1000 total births, respectively).[1] An estimated 68 000 and 76 000 babies per year are born preterm, accounting for 10.5% of births in Malawi and 12% in Zambia.[5] Ranked by increasing rates, Malawi and Zambia were 160th and 155th out of 194 countries for rates of stillbirth in 2015 and 123rd and 127th out of 183 countries for rates of preterm birth in 2014, respectively.[1 5] DIPLOMATIC aims to develop pragmatic clinical trials to test the effectiveness of evidence-based practices and how best to implement them in low-income settings. Consistent and accurate data are required from participating healthcare facilities to efficiently monitor intervention implementation and outcomes, and hence trial results.

Establishing parallel data collection for research projects, rather than integrating this with routine data collection, can lead to a disjointed approach which risks duplication of effort and waste of scarce resources. A learning health system approach has been identified in similar low-income settings, as an effective means of combining data intelligence for quality improvement, with research on the implementation of new interventions and optimising their effectiveness.[6 7] In this context, the learning health system was characterised by a strong stakeholder network, and facilitation of local application of data intelligence, to allow faster integration of evidence-based interventions, and efficient use of the same data to drive research.[7]

### Current perinatal data infrastructure in Malawi

To ensure that the data collection for DIPLOMATIC would integrate with and strengthen existing systems, the collaboration undertook a baseline, in-depth review of the existing data infrastructure relating to perinatal health in Malawi.

Vital events registration in Malawi is mandated by the National Registration Act of 2010, but coverage of birth and death registration is still incomplete, though efforts to improve coverage are in progress.[8 9]

Individuals hold their own medical records as 'health passports' (structured paper-based records), which they present whenever accessing healthcare. For women of childbearing age, the health passport records a summary of medical and obstetric history, contraceptives, antenatal care (ANC) and the delivery record. Neonates receive their own passport at birth. Most district and central hospitals maintain paper-based patient notes when providing inpatient perinatal care. For specialist neonatal care, a standardised admission form is available. A brief note is made in the patients' health passport at discharge,[10] which is usually the main source of information to clinicians on patients' history in case of readmission, as retrieval of patient notes held by healthcare facilities is difficult. Neonatal and maternal records are not routinely linked, and if the mother is not present or deceased, her information is not available for the care of the neonate. An electronic patient record system for ANC is available, but uptake is limited to date (24 out of 85 hospitals and 12 out of 542 health centres across Malawi, as of May 2019).[11] An application for immediate electronic data capture on neonatal admissions (NeoTree) was developed recently and tested in one hospital.[12] This application is designed as a decision support tool, to improve quality of care, and does not have any reporting functionality for integration with local data systems as yet.[12]

In addition to individual patient notes, healthcare facilities maintain 'registers' (individual level paper logs) for clinical care including ANC, maternity/birth record, neonatal resuscitation and Kangaroo Mother Care (KMC), with partial overlap in data items recorded. Aggregate data on healthcare activity are regularly extracted from registers and reported to the Ministry of Health (MoH), via the District Health Information System.[10]

### Importance of high-quality data

Stillbirths, and infant morbidity and mortality associated with preterm births are markers of maternal health and markers of access to high-quality care in pregnancy, especially around the time of childbirth.[13 14] However, global and national estimates of stillbirths and premature births are frequently constrained by data quality.[1 5 15] High-quality data are required to measure the burden of stillbirths and premature births, to assess effectiveness of interventions, and to monitor the benefits of investment to implement evidence-based interventions (or the adverse consequences of a lack of investment). The importance of improving medical records and healthcare information systems, in order to allow monitoring and evaluation, and to improve performance and ultimately outcomes, has also been highlighted as the second of eight WHO standards for improving the quality of maternal and newborn care in health facilities.[16]

### Aim of this work

The aim of this work was to design a minimum dataset, which can be piloted and implemented across all healthcare facilities participating in DIPLOMATIC, to enable electronic collection of consistent and accurate data on perinatal outcomes, risk factors and the coverage of key

care processes and treatments. While the DIPLOMATIC collaboration spans both Zambia and Malawi, this initial work focused on Malawi, with a view to transferring the equivalent data collection, adapted to the Zambian context, in a second stage.

## METHODS

The minimum datasets were drawn up using a multi-modal approach, building on review of the literature and existing data infrastructure for identification of data requirements, followed by consultation with project collaborators and wider local stakeholders in Malawi.[17 18]

### Identification of data requirements

Health passports, neonatal admission forms, and standard registers and the corresponding aggregate data returns to the MoH were reviewed. Variables required for the standard registers and to fulfil aggregate reporting requirements to the MoH were identified for inclusion in the datasets. The resulting list of variables was compared with relevant WHO guidance on ANC,[19] maternal health[20] and newborn health,[21] to ensure that the proposed datasets could be used to assess coverage of recommended clinical best practice. Finally, agreed quality indicators for maternity and neonatal services,[16 22 23] and existing minimum datasets,[24 25] were reviewed to identify additional relevant variables. Wherever possible, international norms and standards were used to underpin definitions.[26]

### Consultation

Identification of the data requirements was followed by iterative rounds of consultation to refine the datasets based on critical evaluation by DIPLOMATIC collaborators and wider stakeholders in Malawi, who provided topic expertise and insights on Malawian health data structure and management.

DIPLOMATIC collaborators were consulted on the appropriateness and feasibility of the draft datasets. In addition to requesting electronic feedback from all collaborators, phone, Skype and email conversations with relevant Malawi-based experts among the collaborators ensured detailed input from obstetricians, paediatricians, midwives, epidemiologists and data scientists. Feedback was recorded against the relevant variables. Where conflicting comments were received, alignment with existing data infrastructure was prioritised, to ensure the data would meet local requirements. The datasets were updated accordingly and shared again with collaborators and discussed at the DIPLOMATIC management group. The draft datasets were shared with relevant professional bodies in Malawi (Nurses and Midwives Council, Association of Obstetricians and Gynaecologists, Association of Paediatricians) facilitated by DIPLOMATIC collaborators who are members of the respective bodies. The draft datasets were also shared with colleagues from the MoH Directorate of Reproductive Health and Directorate of Quality Management and Digital Health. Finally, the

datasets were presented to the Safe Motherhood Technical Working Group, for endorsement by the MoH. The Technical Working Group has multi-sectoral membership including representation from the MoH and the health sector as well as the third sector and academia, and acts as a platform for policy dialogue and coordination of work.

### Patient and public involvement

Patients were not involved in the development of the datasets, but as set out in the DIPLOMATIC stakeholder engagement plan, the extensive experience of local host institutions for the DIPLOMATIC project (University of Malawi College of Medicine; Malawi Liverpool Wellcome Trust, Malawi Epidemiology and Intervention Research Unit) will be used to develop a programme for public awareness of the interventions to be implemented (and to be evaluated using the datasets as described here), which will be supported by the Science Communication team at Malawi Liverpool Wellcome Trust. This will involve consultation with Community Groups, Women's Groups, Patients and Healthcare Workers and will involve multimedia platforms and local meetings.

## RESULTS
### Minimum dataset

Three minimum datasets were drafted to cover ANC, inpatient maternity care (including delivery and care of neonates in maternity settings, and care for miscarriages or complications of miscarriages) and specialist neonatal care. Figure 1 provides a flowchart summarising the datasets' design process. Personal identifiers and demographic variables were included in each of the three datasets to ensure individuals can be uniquely identified and their records linked for care across different settings and at different facilities (horizontal linkage for the same individual during a single pregnancy and subsequent pregnancies, and vertical linkage for mothers and their babies). Malawi is in the process of implementing a unique health identification (UHID) system. Generation of a UHID will be facilitated through registration at the point of care, based on verification of identity using the recently introduced national ID (or birth certification). Use of a Quick Response code on health records for ease and speed of identification of the patient is being considered by the MoH. Each of the three DIPLOMATIC datasets allows recording of the local health record identifier, and this will be used to record the UHID as it embeds into healthcare record use in Malawi. Consistent use of the UHID across records of care will greatly support data linkage and may eventually allow automated imports of demographic details into the patient record, thus reducing data entry effort. However, experience from Scotland shows that linkage of maternal and neonatal health records can remain challenging, even when a unique health identifier is well established. The demographic and ID section of each of the three DIPLOMATIC datasets, is designed to

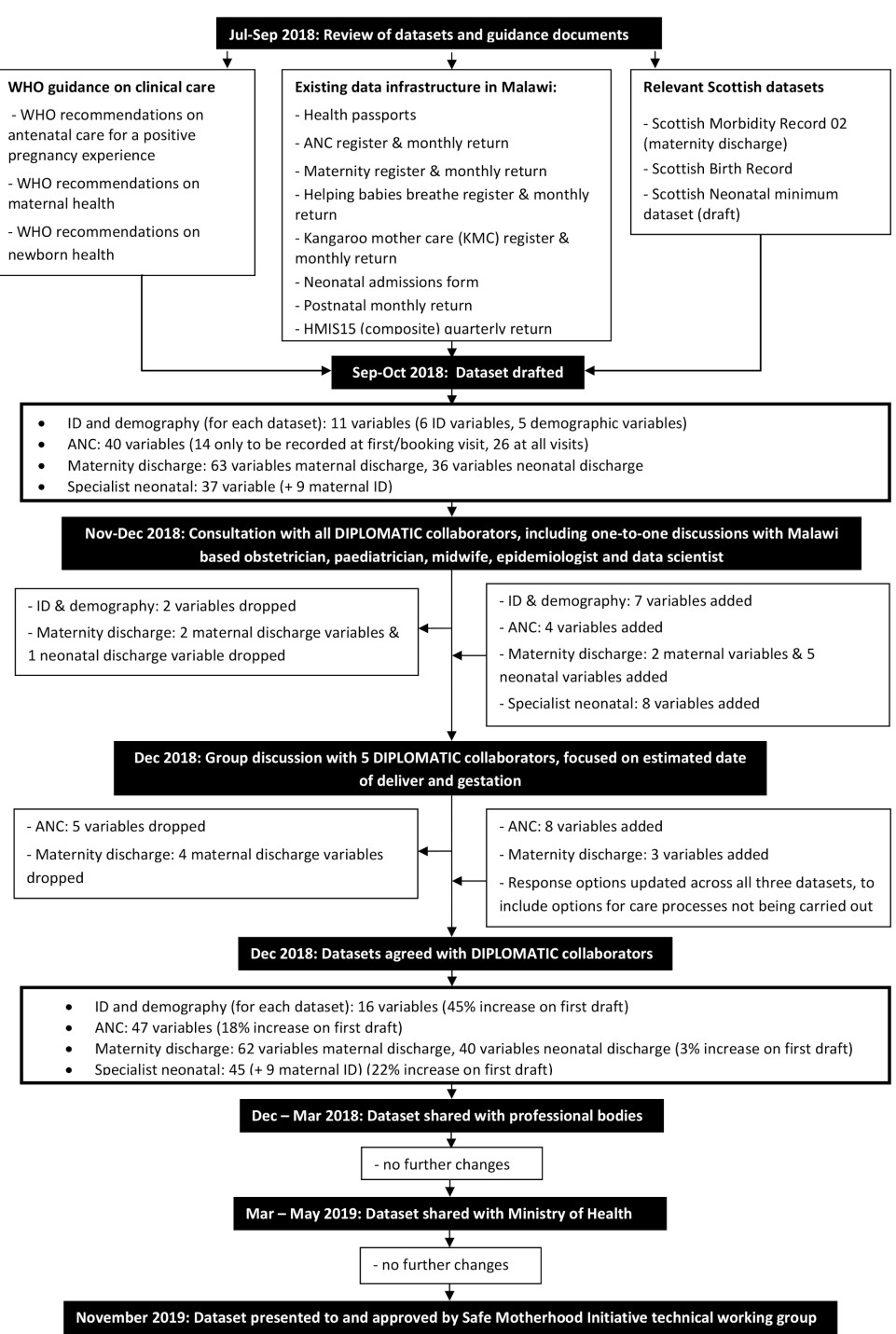

**Figure 1** Flowchart summarising the datasets' design process. ANC, antenatal care; DIPLOMATIC, Using eviDence,Implementation science, and a clinical trial PLatform to Optimise MATernal and newborn health in low Income Countries; ID, identifier.

facilitate data linkage, even in the absence of or in the event of a failure of linkage through a UHID.

To mitigate the risk of incomplete information due to patients receiving some elements of their care in facilities not participating in DIPLOMATIC, the datasets were designed for a hybrid approach between contemporaneous and retrospective data collection. Data would ideally be collected in each of the settings as care is completed (ANC and maternity and neonatal inpatient).

However, the datasets also provide the option to capture retrospective summary information if the relevant data were not captured at the time the care was delivered. The maternity dataset thus includes a summary section on ANC, and the specialist neonatal care dataset allows retrospective recording of summary of maternity/delivery care, if required (figure 2). If the relevant data are captured electronically at the point of care, these retrospective summary sections can be autopopulated.

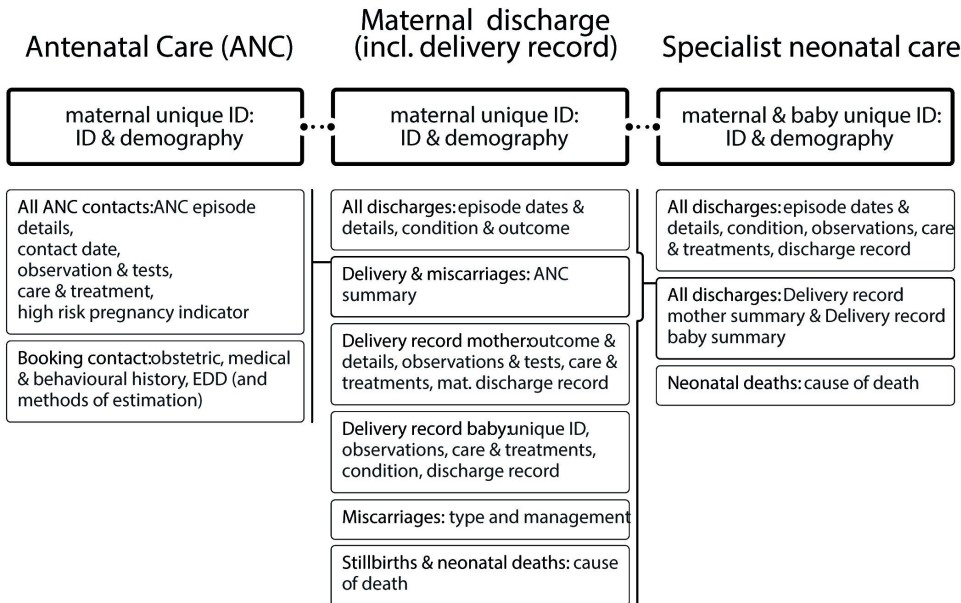

**Figure 2** DIPLOMATIC datasets structure (horizontal lines between the three ID/demography elements highlight that these facilitate the linkage of the different datasets for the same individual and/or mother/baby pairings; different shading within the maternal discharge and the specialist neonatal care datasets show the retrospective summary sections).
**DIPLOMATIC, Using eviDence, Implementation science, and a clinical trial PLatform to Optimise MATernal and newborn health in low Income Countries; EDD, estimated date of delivery; ID, identifier.**

The three datasets (summarised in tables 1–3, available in full on Open Science Framework[17]) included variables to record:

► The main outcomes of interest to the collaboration (stillbirths, preterm births and neonatal deaths), including variables to record estimated date of delivery (EDD), pregnancy outcome, and International Classification of Diseases version 10 coding for cause of death for stillbirths and neonatal deaths where available.

► Key clinical data on potential risk factors and mediators for these outcomes (factors on the causal pathway), including the required variables on obstetric history to derive the Robson criteria (which classify women undergoing caesarean sections into groups defined by a global standard, for comparison of caesarean rates between and within healthcare facilities), underlying maternal conditions of relevance to the current pregnancy, and neonatal observations and underlying conditions.

► Process measures for key aspects of routine healthcare as potential modifiers for these outcomes (factors increasing/reducing risk of adverse outcomes), for ANC, delivery and immediate care of the newborn (including emergency interventions), and specialist neonatal care.

All variables and response options were supported by clear definitions, using internationally recognised norms where possible.[26] Validation cross-checks at data entry were built in to support data quality by design. The datasets were designed to include all required variables to replicate the standard registers used in the respective settings

(ANC, maternity, helping babies breathe (neonatal resuscitation), KMC) and to allow generation of the respective mandatory reports for the MoH via the District Health Information System.

Consultation with the DIPLOMATIC collaborators, wider stakeholders and professional bodies in Malawi resulted in refinement of a number of variables, including additional personal identifiers to create a failsafe for record linkage (based on local experience[27]), and the combination of variables that allow recording of the EDD and the method of estimation. Consultation input also ensured that the response options for all process measures were appropriate for the Malawian healthcare context, including response options to record that items were out of stock. The datasets were endorsed by the MoH through approval by the Safe Motherhood Technical Working Group.

## DISCUSSION

The DIPLOMATIC minimum datasets were designed to cover the maternal and neonatal healthcare journey from ANC to specialist neonatal care. The datasets encompass the required elements to allow precise recording of the relevant outcomes (stillbirths, preterm births and neonatal deaths), as well as risk factors and key care processes. The datasets are aligned with international standards and definitions to ensure comparability of the results, and will enable DIPLOMATIC and the MoH to evaluate implementation and effectiveness of interventions (including care in line with existing policies). They were designed to integrate with and strengthen

**Table 1** Antenatal care (ANC) dataset structure and summary of included variables

| When recorded | Group | Variable | Comment |
|---|---|---|---|
| Booking contact* (cross check ID at each contact) | Unique ID | Maternal unique ID | Record linkage |
| | ID and demography | Maternal names (incl parents' names), date of birth, district and neighbourhood | Record linkage failsafe (building on experience of requirements to uniquely identify individuals from Karonga HDSS) |
| | | Maternal education | Measure of socioeconomic status |
| | | Maternal local ID and local file numbers | Cross-reference local files |
| All ANC contacts | ANC episode details | Facility, level of ANC and care provider | Level of care |
| | Contact date | Contact date | |
| | Observations and tests | Fetal heart rate, symphysis fundal height, any ultrasound scan (USS), maternal haemoglobin, blood pressure, urine protein, infections (syphilis, HIV and hepatitis B status) | For any USS after booking reason is recorded, many of these observations and tests are required for mandatory reporting to MoH |
| | Care and treatment | Tetanus toxoid vaccine doses, malaria prophylaxis, iron and folic acid tablets, preventive anthelmintic; for HIV-positive women: drugs for prevention of mother to child transmission | Many of these process measures are required for mandatory reporting to MoH |
| | High-risk pregnancy indicator | Indicator of whether woman meets any of the WHO high-risk criteria | |
| Booking contact | EDD (and method of estimation) | EDD; USS before or at booking; LMP | EDD is recorded at booking alongside methods used for the estimate (USS, LMP, symphysis fundal height), EDD can subsequently only be altered once—and only on the basis of additional information provided by a first USS occurring after the booking visit |
| | Obstetric, medical and behavioural history | Obstetric history (previous pregnancies, deliveries and c-sections) | Allows derivation of Robson criteria |
| | | Medical history (weight, height, blood group, HIV status, tetanus vaccination history) | |
| | | Behaviour (smoking, alcohol, drugs) | |

*Booking contact: first (booking) visit as captured in the ANC Register, at which a full history is taken and, initial screening for medical, psychological and social risk factors takes places. EDD will also be recorded at the booking visit, although this may subsequently be amended if the first ultrasound scan for the pregnancy takes place after the booking visit.
c-section, caesarean section; EDD, estimated date of delivery; HDSS, Health and Demographic Surveillance System; ID, identifier; LMP, last menstrual period; MoH, Ministry of Health.;

local data systems and meet reporting requirements, and to meet data needs to support decision making and learning for quality improvement in line with the WHO standards for improving the quality of maternal and newborn care in health facilities.[16] Consultation and refinement will need to continue during pilots in the DIPLOMATIC partner facilities, planned to commence in autumn 2020, to ensure the datasets remain aligned with MoH reporting requirements and to enable synergies with projects in the fast moving field of electronic medical record developments, such as an eRegisters platform currently being piloted. In recent discussions the Quality Management and Digital Health Directorate under Malawi's MoH have provided their support to integrate the DIPLOMATIC minimum datasets into this eRegister platform.

Data collection on preterm births and stillbirths is reliant on accurate information on gestation, which underpins the definitions of both concepts. WHO recommends the integration of an ultrasound scan before 24 weeks of gestation into routine ANC, as the most accurate method of determining gestation.[19] At present, ultrasound scans are not part of routine ANC in Malawi, but this was identified as a key evidence-based priority and one of the interventions to be implemented in the healthcare facilities participating in DIPLOMATIC.

**Table 2** Maternal discharge dataset (including delivery record) structure and summary of included variables

| When recorded | Group | Variable | Comments |
|---|---|---|---|
| All discharges | Maternal unique ID | Maternal unique ID | Record linkage |
| | ID and demography | Maternal names (incl parents' names), date of birth, district and neighbourhood | Record linkage failsafe (building on experience of requirements to uniquely identify individuals from Karonga HDSS) |
| | | Maternal education | Measure of socioeconomic status |
| | | Maternal local ID and local file numbers | Cross-reference local files |
| All discharges | Episode dates and details | Admission and discharge dates, facility, admission from, discharge to delivery and miscarriage episodes only: care provider staff group | |
| | Condition and outcome | Reason for admission (category and cause coded in ICD10), condition on discharge | Condition on discharge indicates the outcome that is, miscarriage, delivered or is still pregnant at the end of the care episode |
| Delivery and miscarriage episodes | ANC summary | Obstetric, medical and behavioural history summary | To be autofilled if ANC took place in DIPLOMATIC participating facility, to be manually recorded retrospectively if ANC took place in facility not participating in DIPLOMATIC |
| | | Date of booking contact* | |
| | | Care and treatment summary | |
| | | High-risk pregnancy indicator, multiple pregnancy | |
| | | EDD (and methods of estimation) summary | |
| Delivery record mother | Outcome and details | Number of babies, outcome of pregnancy, birth order, date, time, place, presentation, mode, indications for assisted delivery | |
| | Observation and tests | Partograph used, HIV test, duration of stages of labour | Many of these observations and tests are required for mandatory reporting to MoH |
| | Care and treatments | Preventive/routine: antenatal steroids, magnesium sulfate, labour induction, augmentation of labour, analgesia, uterotonic for active management third stage of labour, episiotomy; companion present; emergency: uterotonic, anticonvulsive, antibiotic, blood transfusion, manual removal of placenta | Many of these process measures are required for mandatory reporting to MoH |
| | Discharge record | Iron and folic acid tablets, preventive anthelminitic, contraception | |
| Delivery record baby | Unique ID | Baby unique ID | |
| | Observations | All: sex, weight; live births: Apgar score, crown-heel length, head circumference | |
| | Care and treatment | Routine: dried and wrapped, delayed cord clamping, cord cut with sterile blade, skin to skin contact initiated, breastfeeding initiated, vitamin K given, tetracycline eye ointment, BCG and OPV, PMTCT initiated (if mother HIV positive); emergency/supportive: resuscitation at birth, systemic antibiotics, Kangaroo Mother Care (KMC) details, other thermal support | Many of these process measures are required for mandatory reporting to MoH |
| | Condition | Congenital malformations, other perinatal complications | |
| | Discharge record | Date, feeding at discharge (mode and content), on KMC at discharge, discharge provider | |
| Miscarriages | Miscarriage type and management | Miscarriage type and management | |

Continued

| When recorded | Group | Variable | Comments |
|---|---|---|---|
| Stillbirths and neonatal deaths | Cause of death | Immediate and underlying cause of death (coded in ICD10) | |

*Booking contact: first (booking) visit as captured in the ANC Register, at which a full history is taken and, initial screening for medical, psychological and social risk factors takes places. EDD will also be recorded at the booking visit, although this may subsequently be amended if the first ultrasound scan for the pregnancy takes place after the booking visit.

ANC, antenatal care; DIPLOMATIC, Using eviDence, Implementation science, and a clinical trial PLatform to Optimise MATernal and newborn health in low Income Countries; EDD, estimated date of delivery; HDSS, Health and Demographic Health Surveillance System; ICD10, International Classification of Diseases version 10; ID, identifier; MoH, Ministry of Health; OPV, oral polio vaccine; PMTCT, prevention of mother to child transmission.

The methods of assessment of gestational age not only determine the precision of estimates for individual pregnancies, but can also lead to systematic differences in population level estimates of prematurity rates,[5] (as exemplified by changes in reporting in the USA[28]). However, most data sources contributing to recent global, regional and national estimates of premature birth rates reported insufficient details on the methods to determine gestational age, to allow the estimates to take this parameter into account.[5] In contrast, the DIPLOMATIC datasets include the relevant parameters to allow recording of whether and when an ultrasound scan was carried out to date the pregnancy, or which other methods were used (last menstrual period, symphysis fundal height), alone or in combination to estimate gestation. These datasets thus allow estimates of preterm birth rates to be adjusted for the methods used to estimate gestational age. The datasets will also allow other important factors to be taken into account, such as multiplicity, and onset of delivery (spontaneous or provider initiated) for analysing both live and total births,[15] and maternal risk factors such as body mass index and hypertensive conditions.[29]

The datasets include process measures on different elements of routine care. These elements serve to meet local reporting requirements, and by recording uptake of evidence-based antenatal, maternal and neonatal care as recommended by WHO,[19–21] they facilitate monitoring of improvements in quality of care. However, each of these data elements also includes response options to reflect that specific care in line with the protocols could not be carried out for example due to stock out of the relevant drugs, in recognition that resource limitations pose important barriers to implementation of care in line with these guidelines. Malawi has adopted the 'Every Newborn Action Plan',[22 30] and is one of the countries participating in the 'Network for Improving Quality of Care for Maternal, Newborn and Child Health', which aims to halve newborn and maternal deaths, and intrapartum stillbirths in participating countries over 5 years.[31] The process measures which will be captured in the DIPLOMATIC datasets are closely aligned to the WHO framework for improving quality of maternal and newborn care in healthcare settings,[16] and will facilitate reporting against a large proportion of the stipulated quality measures. The datasets allow facility-based reporting against 73% (8/11) of core indicators for the framework, and 88% (32/36) of outcome indicators and 77% (44/57) of process/output indicators under the first strategic area of the framework which relates to evidence-based practices for routine care and management of complications.

DIPLOMATIC is staying abreast of the changing electronic health records landscape in Malawi, and is proactively engaging with the Quality Management and Digital Health Directorate under the MoH which is taking the lead to guide the systems to be developed and supported. The collaboration also benefits from the insight of experts, who successfully implemented an electronic data-collection tool in the tertiary care hospital in Blantyre (Surveillance Programme of IN-patients and Epidemiology).[32] However, paper records are currently still an important part of recording and sharing healthcare data in Malawi, and the technical implementation of the DIPLOMATIC minimum datasets will take this into account. The DIPLOMATIC minimum datasets will ideally feed off structured patient records that include the variables of interest, but these may be paper based or electronic, using real-time or retrospective methods of data collection depending on system functionality. Engagement with staff in the participating sites, to discuss advantages of different ways of data capture, to optimise ease of data recording and to establish how this can be supported through existing clerking staff, will be essential to address the challenges of consistent data recording, in the face of pressures on clinical staff time.

Supporting staff to facilitate optimal use of the data will be an important part of piloting and implementation of the datasets. Electronic data can significantly reduce the work required to extract data for audits and patient reviews (compared with extraction from paper records). Experiencing benefits from the data will enhance ownership and buy in from clinical staff, and will be a prerequisite to ensuring sustainability of the datasets. High-quality data providing continuous feedback on quality and safety of care and patient outcomes to healthcare providers can underpin a 'learning health system', which allows data intelligence to foster learning and inform decisions.[33]

**Table 3** Specialist neonatal care dataset structure and summary of included variables

| When recorded | Group | Variable | Use |
|---|---|---|---|
| All discharges | Baby unique ID | Baby unique ID | Record linkage |
| | Baby ID and demography | Baby names, date of birth, sex, district and neighbourhood | Record linkage failsafe |
| | | Baby local ID and local file numbers | Cross-reference local files |
| | Maternal unique ID | Maternal unique ID | Record linkage |
| | Maternal ID and demography | Maternal names (incl parents' names), date of birth, district and neighbourhood | Record linkage failsafe (building on experience of requirements to uniquely identify individuals from Karonga HDSS) |
| | | Maternal education | Measure of socioeconomic status |
| | | Maternal local ID and local file numbers | Cross-reference local files |
| All discharges | Episode dates and details | Admission and discharge dates, facility, admission from, discharge to | |
| | Condition | Reason for admission (category and cause coded in ICD10), | |
| | Observations | Temperature and weight at admission | |
| | Care and treatment | Emergency/supportive: major invasive or therapeutic procedures, Kangaroo Mother Care (KMC) details, other thermal support, respiratory support | Many of these observations and tests are required for mandatory reporting to MoH |
| | Discharge record | Date, weight at discharge, feeding at discharge (mode and content), on KMC at discharge, discharge provider | |
| All discharges | Delivery record mother summary | EDD | To be autofilled if delivery took place in DIPLOMATIC participating facility, to be manually recorded retrospectively if delivery took place in facility not participating in DIPLOMATIC |
| | | Outcome and details | |
| | | Observation and tests | |
| | | Care and treatments | |
| | Delivery record baby summary | Observations | |
| | | Care and treatments | |
| Neonatal deaths | Cause of death | Immediate and underlying cause of death (coded in ICD10) | |

DIPLOMATIC, Using eviDence, Implementation science, and a clinical trial PLatform to Optimise MATernal and newborn health in low Income Countries; EDD, estimated date of delivery; HDSS, Health and Demographic Surveillance System; ICD10, International Classification of Diseases version 10; ID, identifier; MoH, Ministry of Health.;

Data linkage plays an important part in this when transfer of patients across facilities is common. Learning health system approaches have been shown to result in improved data and care quality in limited-resource settings similar to Malawi.[6 7] The DIPLOMATIC datasets have the potential to facilitate a learning health system approach.

### Strengths and limitations

In line with recommended practice, the datasets were designed using a multimodal approach, combining an in-depth review of the contextual information and published as well as grey literature with expert input.[34] Delphi methods to gain expert consensus were considered, but deemed unsuitable due to the large number of variables, many of which form interconnected constructs and could thus not be changed independently of each other. Instead of a Delphi survey, iterative rounds of consultation with DIPLOMATIC collaborators and other stakeholders in Malawi were undertaken. Group discussions were helpful to find agreement on complex constructs, for example, recording of EDD and methods used for its

estimation. While this process also took considerable time to complete, partly due to competing demands for the time of all of the relevant experts, it facilitated agreement on the variables to be prioritised for inclusion in the datasets. Sharing the draft datasets with professional associations and the relevant directorates of the MoH Malawi, and gaining approval by the Safe Motherhood Technical Working Group, was important to ensure there were no objections to the datasets, and to strengthen Malawian ownership, increasing the likelihood that any recommendations arising from future study findings can be implemented and scaled up to increase impact.

One limitation of the DIPLOMATIC datasets is that they are more extensive than other exemplars of national minimum datasets on maternal and neonatal health.[24 25] This poses a challenge for implementation of these datasets in the context of stretched clinical staff, a background of predominantly paper-based data recording and mixed success of electronic data systems.[10] However, it was deemed essential to ensure that the datasets incorporate all data items required for standard registers and can fulfil the Malawian national reporting obligations to the MoH, so the datasets can be embedded in existing systems and strengthen these (in line with the 'record once, use often' principle). This approach avoided creation of parallel datasets specific only to the research data needs, which would have been smaller, but would have overlapped to a large extent with existing paper-based registers, leading to duplication of effort and no benefit to the existing system.

The DIPLOMATIC collaboration is focused on healthcare interventions, and the minimum datasets can only capture maternal and newborn care in health facilities. Home births and complications associated with these, or stillbirths out with health facilities, would only be captured if aftercare was received in a health facility, or if recorded as part of a subsequent healthcare episode.

## CONCLUSIONS

By combining contextual work with expert input, it was possible to design datasets, which build on existing structures, and can support local reporting requirements, as well as meeting the research data needs of DIPLOMATIC. The work to design the datasets highlights the need for a collaborative approach to ensure synergies and to strengthen existing systems, and avoid duplication of data and effort. Consultation on data collection is also essential to ensure local ownership of the data and any findings arising from it. The datasets have the potential to underpin a 'learning health system' approach, and their piloting and implementation will be aimed at supporting quality of care to improve outcomes. Implementation of these datasets could thus contribute to fulfilling the aims of the Network for Improving Quality of Care for Maternal, Newborn and Child Health, in particular to strengthening of the health information system, 'to enable early, appropriate action to improve the care of every woman and newborn' (standard 2).[16] The datasets may also serve beyond the use by facilities participating in DIPLOMATIC, most immediately as an exemplar on how to incorporate methods of gestation estimates into routine data collection, in settings where access to early pregnancy ultrasound is limited and different methods are likely to be in use. In a wider context they also serve as a transferable example of strengthening data infrastructure and use of robust data, to underpin a learning healthcare system approach in low-income settings.

**Author affiliations**
[1]Information Services Division, NHS National Services Scotland, Edinburgh, UK
[2]Public Health, NHS Greater Glasgow and Clyde, Glasgow, UK
[3]MRC Centre for Reproductive Health, The University of Edinburgh, Edinburgh, UK
[4]Department of Obstetrics and Gynaecology, University of Malawi College of Medicine, Blantyre, Malawi
[5]Paediatric Department, University of Malawi College of Medicine, Blantyre, Malawi
[6]Malawi-Liverpool-Wellcome Trust Clinical Research Programme, Blantyre, Malawi
[7]Faculty of Midwifery, Neonatal and Reproductive Health Studies, University of Malawi Kamuzu College of Nursing, Blantyre, Malawi
[8]Usher Institute of Population Health Sciences and Informatics, The University of Edinburgh, Edinburgh, UK
[9]Queen's Medical Research Institute, The University of Edinburgh Centre for Cardiovascular Science, Edinburgh, UK
[10]Edinburgh Clinical Trials Unit, The University of Edinburgh Usher Institute of Population Health Sciences and Informatics, Edinburgh, UK
[11]Malawi Epidemiology and Intervention Research Unit, Lilongwe, Malawi
[12]Institute of Infection and Global Health, University of Liverpool, Liverpool, UK
[13]Centre for Global Health Research, The University of Edinburgh Usher Institute of Population Health Sciences and Informatics, Edinburgh, UK
[14]Faculty of Health Sciences, University of Bristol, Bristol, UK
[15]Salvesen Mindroom Research Centre, University of Edinburgh, Edinburgh, UK

**Acknowledgements** This work would not have been possible without consultation with stakeholders from the Ministry of Health Malawi, the Malawian Nurses and Midwives Council, the Malawian Association of Obstetricians and Gynaecologists, and the Malawian Association of Paediatricians. We would like to thank Jakub Adamów for producing the graphic for this paper.

**Collaborators** The DIPLOMATIC collaborators: Catherine Bamuya, Esmie Banda, James Boardman, Effie Chipeta, Mia Crampin, Sarah Cunningham Burley, Jean Desire Kabamba, Elizabeth Grant, Caroline Hollins Martin, Aisha Holloway, Khondwhani Kawaza, Doris Kayambo, Corrine Love, Brian Magowan, Monica Malata, Martha Masamba, Fadhila Mazanderani, Gladys Membe-Gadama, Catherine Mkandawire, Patricia Munthali, Peter Mwaba, Shakira Namisengo, Everist Njelesani, Linda Nyondo-Mipando, Hilary Pinnock, Muriel Syacumpi, Frank Taulo and Alexandra Viner.

**Contributors** BvW and RW designed the protocol for the development of the datasets and drafted the first version of the datasets. BvW wrote the first draft of the manuscript. BvW, EW, DW and RW synthesised the information needed to identify the dataset requirements. LAG, QD, CM and EC provided expert input on the dataset requirements, and current data infrastructure and management in Malawi. LAG, QD and EC led on engagement with the professional bodies. All authors (BvW, EW, DW, LAG, QD, CM, EC, SJS, RR, JN, EM, BF, HC, JEN and RW) provided significant intellectual input to the iterative development of the datasets. All authors (BvW, EW, DW, LAG, QD, CM, EC, SJS, RR, JN, EM, BF, HC, JEN and RW) provided input to drafting the manuscript.

**Funding** This research was funded by the National Institute for Health Research (NIHR) (GHR Project: 17/63/08 DIPLOMATIC collaboration) using UK aid from the UK Government to support global health research. The views expressed in this publication are those of the authors and not necessarily those of the NIHR or the Department of Health and Social Care. SJS is funded by a Wellcome Trust Clinical Career Development Fellowship (209560).

**Competing interests** SJS reports grants from NIHR HTA, non-financial support from HOLOGIC, non-financial support from PARSAGEN, non-financial support from

MEDIX BIOCHEMICA, during the conduct of the study; and SJS declares being a member of the HTA general committee. JN reports membership of the following NIHR boards: Commissioning Priority Review decision-making committee (2015); Health Technology Assessment (HTA) Commissioning Board (2010–2016); HTA Commissioning Sub-Board (Expression of Interest) (2014); HTA Funding Boards Policy Group (2016–2019); HTA General Board (2016–2019); HTA Post-Board funding teleconference (2016–2019); NIHR Clinical Trials Unit Standing Advisory Committee (2018–present); NIHR HTA and Efficacy and Mechanism Evaluation Editorial Board (2014–2019); Pre-exposure Prophylaxis Impact Review Panel (2017). JEN reports being named as Principal Investigator on government and charitable research grants to her institution which aim to improve pregnancy outcome. In the last 3 years, she has provided consultancy to Pharma companies GSK and Dilafor: her institution was remunerated for this. Her institution has received travel and subsistence expenses from Merck to facilitate her speaking at a Merck-sponsored symposium on metformin. She is on Subpanel A1 for REF, and on a Wellcome Trust Science interview panel, and receive personal remuneration for each.

**Patient consent for publication** Not required.

**Provenance and peer review** Not commissioned; externally peer reviewed.

**Data availability statement** Dataset specifications are available in a public, open access repository: The full specifications of the datasets, the development of which is described in this manuscript, are available on Open Science Framework: von Wissmann B, Gadama L, Dube Q *et al*. The DIPLOMATIC Collaboration, Work Package 2: Core datasets. 2020. https://osf.io/d3s7j.

**ORCID iDs**
Beatrix von Wissmann http://orcid.org/0000-0002-0980-4545
Donald Waters http://orcid.org/0000-0002-5169-8539
Sarah Jane Stock http://orcid.org/0000-0003-4308-856X
Jane E Norman http://orcid.org/0000-0001-6031-6953
Rachael Wood http://orcid.org/0000-0003-4453-623X

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
