## [Reviewer comments · BMJ Open]

ARTICLE DETAILS

TITLE (PROVISIONAL)	Informing prevention of stillbirth and preterm birth in Malawi: development of a minimum dataset for health facilities participating in the DIPLOMATIC collaboration
AUTHORS	von Wissmann, Beatrix; Wastnedge, Elizabeth; Waters, Donald; Gadama, Luis; Dube, Queen; Masesa, Clemens; Chodzaza, Elizabeth; Stock, Sarah; Reynolds, Rebecca; Norrie, John; Makwakwa, Enita; Freyne, Bridget; Campbell, Harry; Norman, Jane; Wood, Rachael

VERSION 1 – REVIEW

REVIEWER	Deepa Dongarwar Baylor College of Medicine, Houston, TX
REVIEW RETURNED	10-May-2020

GENERAL COMMENTS	This paper aims to design a minimum dataset, to be piloted and implemented across all healthcare facilities participating in DIPLOMATIC, but starting with health centers in Malawi; and to enable electronic collection of consistent and accurate data on perinatal outcomes, risk factors and the coverage of key care The title seems odd. Should consider 'Reporting stillbirth and preterm birth in Malawi: development of a minimum dataset for health facilities participating in the DIPLOMATIC collaboration' Abstract line 10: What do the authors mean by 'An early objective'? Abstract line 22-25: Sentence doesn't read well. What about the 'the existing local data infrastructure and reporting requirements.'? Please re-phrase Abstract: Methods: The first and 2nd line need to be joined by what the purpose of the study was; i.e. to identify data elements that will be a part of the minimum dataset Abstract results: should include how many databases are being created and what do they contain. Introduction : Consider adding more recent information on global stillbirth in line 1 of introduction. Introduction 1st para, last line: needs proper punctuations - Babies born preterm, who survive, remain at risk of developmental delay,[3] and long term physical morbidities. Introduction 2nd paragraph: information on Zambia and Malawi is pretty outdated. Consider adding more recent details. Introduction page 5 line 45: Grammatical error. Change to "baseline, in-depth review" Page 6, line 35-39 – When talking about "Stillbirths, and morbidity and mortality", whose mortality are you talking about? Child's or mother's? Please specify. Figure 1 is not at all legible. The colors are all merged and the orientation is messed up. Better quality image needs to be provided Table 1: You mention 'Karonga DHSS' in table 1, while in footnote it
--

	is HDSS. Which one is it? Methods: When was this project started? How much time did it take for each iteration and to complete the whole process? Should include these details Methods: Details on how many collaborators were included in the dataset designing process is required. How many rounds of iterative variable selection process was followed till the final design was finalized? How are the home-births, complications of home births, stillbirths which were not attended by doctors captured? Is the minimum dataset already in use? Provide some details on how and when it is going to be used around in Malawi.
--	---

REVIEWER	Gerald Douglas University of Pittsburgh, USA. Global Health Informatics Institute, Malawi.
REVIEW RETURNED	16-May-2020

GENERAL COMMENTS	I think this is vitally important work and would like to thank the co-authors for their efforts thus far. I provide a summary of my key feedback point below (additional and more granular feedback provided in the attached document).  • The QI framework, grounded in the context of a “learning health system”, could be more prominent in the manuscript (LHS not mentioned until last para of manuscript). Suggest defining LHS and discussing in the introduction. • The QI cycle (PDSA) is a systematic set of processes that all need to work if QI is to be achieved (a chain is as strong as it weakest link). In order for data to impact quality the data has to be collected and put in a form for analysis that assures completeness, accuracy and timeliness. The size of the data set and the mode of capture and entry into electronic form for analysis will have a significant impact on data quality. An excessively large data set would compromise the entire process. Discussing the data set without presenting details on how data capture and entry into electronic form can be feasibly achieved in the context of a Malawian health facility does not provide an adequate overall picture. There is mention of real-time and retrospective data entry. This could be elaborated on for greater clarity. • There is some discussion about the importance of uniquely identifying patients. However, there is no mention of how the recently introduced national ID might be used. Describing how your approach builds on a national system for identification could add credibility. • Data collection is time-consuming. Malawi has a crisis in human resources for health. “Routine” data collection implies this is done consistently. However, there is significant evidence of incomplete data from these “routine” systems. Not discussing how you plan to address this issue undermines the overall credibility of this broader QI approach. • The protocol is described in the methods section but the results section lacks detail. More quantitative results can inform an understanding of the degree to which the size of the data set changed (increased by 10% vs doubled in size). What was the size of the data sets you started with? When comparing the WHO data set with the existing Malawi data set how much intersection was there? Was one a subset of the other? A Venn diagram for each night be useful. As you iteratively “refined” each data set through consultation, how did this affect the size of the data set
---

	(additions/subtractions/modifications)? A graphic (flow diagram) showing changes after each step could be useful. At the end of this process, how many data elements were in each data set compared to the initial data sets routinely used in Malawi? Something along the lines of this graphic https://www.researchgate.net/figure/Flow-chart-outlining-selection-of-papers-for-review_fig1_256836094.  • Reference to consultation with Ministry of Health should include specific relevant bodies within MoH. The RHU and QMD are mentioned. To what degree were other relevant bodies within MoH consulted (e.g. Expanded Programme for Immunization (vaccination relevance) and Department of HIV and AIDS (PMCTC relevance)?
--	--

VERSION 1 – AUTHOR RESPONSE

Reviewer: 1

Reviewer Name: Deepa Dongarwar

Institution and Country: Baylor College of Medicine, Houston, TX

Please state any competing interests or state 'None declared': None

Please leave your comments for the authors below

Attachment: Review comments.pdf

The comments in the PDF were identical to these in the cover email and are thus only listed once.

This paper aims to design a minimum dataset, to be piloted and implemented across all healthcare facilities participating in DIPLOMATIC, but starting with health centers in Malawi; and to enable electronic collection of consistent and accurate data on perinatal outcomes, risk factors and the coverage of key care.

The title seems odd. Should consider 'Reporting stillbirth and preterm birth in Malawi: development of a minimum dataset for health facilities participating in the DIPLOMATIC collaboration'

We thank the reviewer for their suggestion, but would like to retain the original title, to reflect that the minimum datasets were developed not just to assist reporting, but to inform practice and facilitate quality improvement as well as implementation science research to prevent stillbirths and preterm births.

Abstract line 10: What do the authors mean by 'An early objective'?

We were referring to the design of the datasets being an objective for the early phase of the project; however this phrase has been dropped in the restructuring of the abstract to meet the editorial requirements.

Abstract line 22-25: Sentence doesn't read well. What about the 'the existing local data infrastructure and reporting requirements.'? Please re-phrase

In line with the suggestion, this sentence has now been combined with the following sentence, as follows:

'They have been informed by extensive consultation and are designed to integrate with and strengthen existing local data infrastructure and reporting as well as meeting research data needs.'

Abstract: Methods: The first and 2nd line need to be joined by what the purpose of the study was; i.e. to identify data elements that will be a part of the minimum dataset

The first sentence of the methods section (now under heading design) has been edited to include the purpose of the study, in line with the reviewer's suggestion, as follows:

'Published and grey literature was reviewed alongside extensive in-country consultation to define relevant clinical best practice guidance, and the existing local data and reporting infrastructure, to identify requirements for the minimum datasets.'

Abstract results: should include how many databases are being created and what do they contain.
The results section has been edited, in line with the reviewer's suggestion, to specify the number of minimum datasets, and which setting they are for (antenatal, maternity and specialist neonatal care), as well as what they contain (data elements recording outcomes, risk factors and key care processes). It now reads:

'The resulting three minimum datasets cover the maternal and neonatal healthcare journey for antenatal, maternity and specialist neonatal care, with provision for effective linkage of records for mother/baby pairs. They can facilitate consistent, precise recording of relevant outcomes (stillbirths, preterm births, neonatal deaths), risk factors and key care processes.'

Introduction : Consider adding more recent information on global stillbirth in line 1 of introduction.
We revisited the literature, but found that the papers cited are still the most recently published global estimates currently available, as also cited on the relevant WHO web pages.

Introduction 1st para, last line: needs proper punctuations - Babies born preterm, who survive, remain at risk of developmental delay,[3] and long term physical morbidities.
We corrected the punctuation, in line with the reviewer's comment.

Introduction 2nd paragraph: information on Zambia and Malawi is pretty outdated. Consider adding more recent details.
We revisited the literature, but found that the papers cited are still the most recently published global estimates currently available. We wanted to use global estimates to allow methodologically consistent comparison of estimates for Malawi and Zambia with other countries.

Introduction page 5 line 45: Grammatical error. Change to "baseline, in-depth review"
We corrected the sentence, in line with the reviewer's comment.

Page 6, line 35-39 – When talking about "Stillbirths, and morbidity and mortality", whose mortality are you talking about? Child's or mother's? Please specify.
This refers to morbidity and mortality of the child and has been specified as requested:

'Stillbirths, and infant morbidity and mortality associated with preterm births are markers of maternal health and markers of access to high quality care in pregnancy, especially around the time of childbirth.'

Figure 1 is not at all legible. The colors are all merged and the orientation is messed up. Better quality image needs to be provided
We apologise that the figure was distorted when the proof was created for the review process. A revised version is being submitted with the updated manuscript (now Figure 2).

Table 1: You mention 'Karonga DHSS' in table 1, while in footnote it is HDSS. Which one is it?
We thank the reviewer for alerting us to this inconsistency, the full name is Karonga Health and Demographic Surveillance System, and the abbreviation in table 1 has been updated to HDSS, in line with the rest of the manuscript.

Methods: When was this project started? How much time did it take for each iteration and to complete the whole process? Should include these details.
We have created a flowchart (now Figure 1), which includes these details.

Methods: Details on how many collaborators were included in the dataset designing process is required. How many rounds of iterative variable selection process was followed till the final design was finalized?
We have created a flowchart (now Figure 1), which includes these details.

How are the home-births, complications of home births, stillbirths which were not attended by doctors captured?

The DIPLOMATIC collaboration is focussed on health care interventions, and the minimum datasets can only capture maternal and newborn care in health facilities. Home births and complications associated with these, or stillbirths out with health facilities, would only be captured if aftercare was received in a health facility, or if recorded as part of a subsequent health care episode.

The above clarification has been integrated into the strengths and limitations section of the discussion.

The majority of women in Malawi deliver in health facilities. The proportion of deliveries, that occurred in institution increased from 55% in 1992 to 91% in 2015-16, as shown by the Malawi Demographic and Health Survey 2015-2016. This shift reflects that delivering in a facility was made compulsory in Malawi, and that traditional birth attendants were banned for a time and that authorities imposed fines on women delivering outside facilities.

Through close collaboration with Health and Demographic Surveillance Systems, comparison with outcomes and complications for homebirths, and stillbirths occurring in the community may be facilitated in future, however this is out with the scope of the work presented here.

Is the minimum dataset already in use? Provide some details on how and when it is going to be used around in Malawi.

The minimum datasets are intended for implementation in DIPLOMATIC partner facilities in the first instance. A change of the Electronic Medical Records System implementation partner of the Ministry of Health, and challenges of the COVID19 response have led to delays. In recent discussions the Quality Management and Digital Health (QMDH) Directorate under Malawi's Ministry of Health have provided their support to integrate the DIPLOMATIC minimal datasets into the eRegister developed on OpenSRP (Open Smart Reporting Platform).

QMDH is planning to use this eRegister platform to integrate all systems for primary health care. Currently the platform is underdevelopment and piloting is underway in 1 health facility in southern Malawi. This work is expected to be completed in July – August 2020, and piloting in DIPLOMATIC partner facilities is now planned for autumn 2020, coronavirus permitting. This has now been clarified at the end of the first paragraph of the discussion, which now states:

'Consultation and refinement will need to continue during pilots in the DIPLOMATIC partner facilities, planned to commence in autumn 2020, to ensure the datasets remain aligned with MoH reporting requirements and to enable synergies with projects in the fast moving field of electronic medical record (EMR) developments, such as an eRegisters platform currently being piloted. In recent discussions the Quality Management and Digital Health (QMDH) Directorate under Malawi's MoH have provided their support to integrate the DIPLOMATIC minimal datasets into this eRegister platform.'

Reviewer: 2

Reviewer Name: Gerald Douglas

Institution and Country:

University of Pittsburgh, USA.

Global Health Informatics Institute, Malawi.

Please state any competing interests or state 'None declared': None.

Please leave your comments for the authors below

Attachment: BMJ_review_DIPLOMATIC_gdouglas.pdf

I think this is vitally important work and would like to thank the co-authors for their efforts thus far.
We would like to thank the reviewer for this assessment of the importance of this work.

I provide a summary of my key feedback point below (additional and more granular feedback provided in the attached document).

- The QI framework, grounded in the context of a “learning health system”, could be more prominent in the manuscript (LHS not mentioned until last para of manuscript). Suggest defining LHS and discussing in the introduction.

In line with this suggestion, we have included an additional paragraph in the second section of the introduction, which outlines the concept and advantages of a learning health system:

‘Establishing parallel data collection for research projects, rather than integrating this with routine data collection, can lead to a disjointed approach which risks duplication of effort and waste of scarce resources. A learning health system approach has been identified in similar low income settings, as an effective means of combining data intelligence for quality improvement, with research on the implementation of new interventions and optimising their effectiveness.[6,7] In this context, the learning health system was characterised by a strong stakeholder network, and facilitation of local application of data intelligence, to allow faster integration of evidence based interventions, and efficient use of the same data to drive research. [7]’

- The QI cycle (PDSA) is a systematic set of processes that all need to work if QI is to be achieved (a chain is as strong as its weakest link). In order for data to impact quality the data has to be collected and put in a form for analysis that assures completeness, accuracy and timeliness. The size of the data set and the mode of capture and entry into electronic form for analysis will have a significant impact on data quality. An excessively large data set would compromise the entire process. Discussing the data set without presenting details on how data capture and entry into electronic form can be feasibly achieved in the context of a Malawian health facility does not provide an adequate overall picture. There is mention of real-time and retrospective data entry. This could be elaborated on for greater clarity.

We agree with the reviewer regarding the challenges posed by an extensive dataset. This is highlighted in the strengths and limitations sections (summary section, and within the discussion), which have been rephrased to identify the size of the dataset more clearly as a limitation.

In the summary section:

- *One limitation of this inclusive approach is that the datasets are more extensive than other exemplars of national minimum datasets.*
- *However the approach ensures that the datasets can integrate with existing systems and meet local data requirements, and also have the capacity to evaluate other Ministry of Health initiatives, rather than addressing research requirements only.*

Within the discussion:

‘One limitation of the DIPLOMATIC datasets is that they are more extensive than other exemplars of national minimum datasets on maternal and neonatal health.[24, 25] This poses a challenge for implementation of these datasets in the context of stretched clinical staff, a background of predominantly paper based data recording and mixed success of electronic data systems.[10]’

An additional paragraph has been added to the discussion, elaborating on plans for developing an appropriate system for data capture:

‘DIPLOMATIC is staying abreast of the changing electronic health records landscape in Malawi, and is proactively engaging with the Quality Management and Digital Health Directorate under the MoH which is taking the lead to guide the systems to be

development and supported. The collaboration also benefits from the insight of experts, who successfully implemented an electronic data-collection tool in the tertiary care hospital in Blantyre (Surveillance Programme of IN-patients and Epidemiology –SPINE).[32] However, paper records are currently still an important part of recording and sharing health care data in Malawi, and the technical implementation of the DIPLOMATIC minimum datasets will take this into account. The DIPLOMATIC minimum datasets will ideally feed off structured patient records that include the variables of interest, but these may be paper based or electronic, using real-time or retrospective methods of data collection depending on system functionality. Engagement with staff in the participating sites, to discuss advantages of different ways of data capture, to optimise ease of data recording, and to establish how this can be supported through existing clerking staff, will be essential to address the challenges of consistent data recording, in the face of pressures on clinical staff time.'

In the results section, the explanation on contemporaneous and retrospective data collection has been expanded as follows:

'To mitigate the risk of incomplete information due to patients receiving some elements of their care in facilities not participating in DIPLOMATIC, the datasets were designed for a hybrid approach between contemporaneous and retrospective data collection. Data would ideally be collected in each of the settings as care is completed (ANC and maternity and neonatal inpatient). However, the datasets also provide the option to capture retrospective summary information if the relevant data was not captured at the time the care was delivered. The maternity dataset thus includes a summary section on ANC, and the specialist neonatal care dataset allows retrospective recording of a summary of maternity/delivery care, if required (Figure 1). If the relevant data is captured electronically at the point of care, these retrospective summary sections can be auto-populated.'

- There is some discussion about the importance of uniquely identifying patients. However, there is no mention of how the recently introduced national ID might be used. Describing how your approach builds on a national system for identification could add credibility.

We have now elaborated on this in the results section of the manuscript as follows:

'Malawi is in the process of implementing a unique health identification (UHID) system. Generation of a UHID will be facilitated through registration at the point of care, based on verification of identity using the recently introduced national ID (or birth certification for those under 16 years). Use of a Quick Response (QR) code for ease and speed of identification of the patient is being considered by the MoH. Each of the three DIPLOMATIC datasets allows recording of the local health record identifier, and this will be used to record the UHID as it embeds into health care record use in Malawi. Consistent use of the UHID across records of care will greatly support data linkage and may eventually allow automated imports of demographic details into the patient record, thus reducing data entry effort. However experience from Scotland shows that linkage of maternal and neonatal health records can remain challenging, even when a unique health identifier is well established. The demographic and ID section of each of the three DIPLOMATIC datasets, is designed to facilitate data linkage, even in the absence of or in the event of a failure of linkage through a UHID.'

- Data collection is time-consuming. Malawi has a crisis in human resources for health. "Routine" data collection implies this is done consistently. However, there is significant evidence of incomplete data from these "routine" systems. Not discussing how you plan to address this issue undermines the overall credibility of this broader QI approach.

We agree with this observation, and have expanded on plans in the discussion section:

'DIPLOMATIC is staying abreast of the changing electronic health records landscape in Malawi, and is proactively engaging with the Quality Management and Digital Health Directorate under the MoH which is taking the lead to guide the systems to be development

and supported. The collaboration also benefits from the insight of experts, who successfully implemented an electronic data-collection tool in the tertiary care hospital in Blantyre (Surveillance Programme of IN-patients and Epidemiology – SPINE).[32] However, paper records are currently still an important part of recording and sharing health care data in Maawi, and the technical implementation of the DIPLOMATIC minimum datasets will take this into account. The DIPLOMATIC minimum datasets will ideally feed off structured patient records that include the variables of interest, but these may be paper based or electronic, using real-time or retrospective methods of data collection depending on system functionality. Engagement with staff in the participating sites, to discuss advantages of different ways of data capture, to optimise ease of data recording, and to establish how this can be supported through existing clerking staff, will be essential to address the challenges of consistent data recording, in the face of pressures on clinical staff time.'

- The protocol is described in the methods section but the results section lacks detail. More quantitative results can inform an understanding of the degree to which the size of the data set changed (increased by 10% vs doubled in size).

- What was the size of the data sets you started with?

- When comparing the WHO data set with the existing Malawi data set how much intersection was there? Was one a subset of the other? A Venn diagram for each might be useful.

- As you iteratively “refined” each data set through consultation, how did this affect the size of the data set (additions/subtractions/modifications)? A graphic (flow diagram) showing changes after each step could be useful. At the end of this process, how many data elements were in each data set compared to the initial data sets routinely used in Malawi? Something along the lines of this graphic https://www.researchgate.net/figure/Flow-chart-outlining-selection-of-papers-for-review_fig1_256836094.

We have created a flow diagram (now Figure 1) on the development of the datasets, as suggested, and have quantified the proportion of the WHO framework for improving quality of maternal and newborn care in healthcare settings in the text.

The process measures which will be captured in the DIPLOMATIC datasets are closely aligned to the WHO framework for improving quality of maternal and newborn care in healthcare settings,[16] and will facilitate reporting against a large proportion of the stipulated quality measures. The datasets allow facility based reporting against 73% (8/11) of core indicators for the framework, and 88% (32/36) of outcome indicators and 77% (44/57) of process/output indicators under the first strategic area of the framework, which relates to evidence based practices for routine care and management of complications.

- Reference to consultation with Ministry of Health should include specific relevant bodies within MoH. The RHU and QMD are mentioned. To what degree were other relevant bodies within MoH consulted (e.g. Expanded Programme for Immunization (vaccination relevance) and Department of HIV and AIDS (PMCTC relevance)?

The methods section specifies the relevant units and departments within the Ministry of Health, with whom consultation took place. We concur that several elements of the data to be collected are relevant for other cross cutting programmes, such as the Expanded Programme for Immunization and the Department for HIV and AIDS. However, the relevant variables were adopted into the DIPLOMATIC minimum datasets unchanged from current Ministry of Health specifications for registers and monthly returns for antenatal, maternity and neonatal care, and thus no specific consultation with these programmes/ departments was carried out. We recognise that ongoing close working with the Ministry of Health will be vital for the success of the collaboration, and are closely following the advice of the Reproductive Health Directorate and the Quality Management and Digital Health Directorate on the appropriate engagement.

Reviewer 2 Detailed comments from PDF

We would like to thank the reviewer for taking the time to provide these detailed comments, and for sharing their insights. We wanted to provide full responses to each comment, rather than cross referring between comments. Thus many of our responses to these detailed comments include replication of the responses also provided under the key summary above (as well as extensions and additions to fully address the reviewer's comments).

Informing prevention of stillbirth and preterm birth in Malawi: development of a minimum dataset for health facilities participating in the DIPLOMATIC collaboration

Reviewer feedback. 16 May 2020

Quality improvement cycles rely on good data

I think this message could be stressed earlier in the manuscript. The phrase “learning health system” is only discussed in the last 5 lines of the manuscript. Concepts of a PDSA cycle are mentioned earlier ...

“DIPLOMATIC aims to develop pragmatic clinical trials to test the effectiveness of evidence based practices and how best to implement them in low income settings. Consistent and accurate data are required from participating healthcare facilities to efficiently monitor intervention implementation and outcomes and hence trial results.” I think this notion of the learning health system would fit well in the introduction. Applying evidence-based medicine approaches at the health system level is slowly gaining momentum, and I believe this manuscript could frame the work solidly in that context.

In line with this suggestion, we have included an additional paragraph in the second section of the introduction, which outlines the concept and advantages of the learning health system approach:

'Establishing parallel data collection for research projects, rather than integrating this with routine data collection, can lead to a disjointed approach which risks duplication of effort and waste of scarce resources. A learning health system approach has been identified in similar low income settings, as an effective means of combining data intelligence for performance monitoring and quality improvement, with research on the implementation of new interventions and optimising their effectiveness.[6,7] In this context, the learning health system was characterised by a strong stakeholder network, and facilitation of local application of data intelligence, to allow faster integration of evidence based interventions, and efficient use of the same data to drive research.[7]'

Taking a “systems thinking” approach to better data

This manuscript addresses an important problem. Getting better data requires consideration for who will document the event that leads to the data, how and when, additionally, who will enter the data into a computer for analysis, how and when. This “system” has to be robust. Selecting approaches that will result in high data quality depend on how much data is being collected. There is no question that we need better data to measure the baseline and to detect changes in outcomes (positive and negative) based on interventions. Identifying the optimal dataset is the first step. This is difficult, and in this manuscript the team has taken a systematic approach that is grounded in a philosophy of inclusion, both in terms of data elements and stakeholders. However, the implication of inclusion is that the number of data elements will increase (the “new” routine). Given finite and limited staff to record data the implication for data quality (defined here as completeness and accuracy) is that as the number of data elements goes up, the quality of the data goes down. One personal example of this was with the dataset form HIV VCT for Malawi AIDS Counseling and Resources Organization in 2003. Counselors were asked to document roughly 120 data elements for each client, of which one section containing up to 28 data elements comprising 7 data points for each sexual partner over the past 6 months, not to exceed 4 partners. Some counselors reported that the burden required to capture all that data was too much and that they omitted data when they had many clients. At that time the process was all on paper, so this could not be attributed to technology challenges (those would come later). Data collection and data entry are time consuming and error prone. References were made to contemporaneous (real-time) data collection over retrospective. In the statement “Data would ideally be *collected* contemporaneously in each of the settings (ANC and maternity and neonatal inpatient) as care is completed.” could be interpreted as collected on a paper form for subsequent

entry into a computer or alternatively entered into the computer at that time. It would be helpful to elaborate on this in the manuscript. Work at Bwaila maternity hospital starting 2010 where data was captured in real time was fraught with problems, some of which related to the parallel use of the computer system and government registers. Given the high level of consultation that the DIPLOMATIC team has done with MoH, one hopes that any requirement for parallel use of data collection tools has been negotiated.

We agree with the reviewer regarding the challenges posed by an extensive dataset. This is highlighted in the strengths and limitations sections (summary section, and within the discussion), which have been rephrased to identify the size of the dataset more clearly as a limitation.

In the summary section:

- *One limitation of this inclusive approach is that the datasets are more extensive than other exemplars of national minimum datasets.*
- *However the approach ensures that the datasets can integrate with existing systems and meet local data requirements, and also have the capacity to evaluate other Ministry of Health initiatives, rather than addressing research requirements only.*

Within the discussion:

'One limitation of the DIPLOMATIC datasets is that they are more extensive than other exemplars of national minimum datasets on maternal and neonatal health.[24, 25] This poses a challenge for implementation of these datasets in the context of stretched clinical staff, a background of predominantly paper based data recording and mixed success of electronic data systems.[10]

An additional paragraph has been added to the discussion, elaborating on plans for developing an appropriate system for data capture, recognising the challenges that this will pose:

'DIPLOMATIC is staying abreast of the changing electronic health records landscape in Malawi, and is proactively engaging with the Quality Management and Digital Health Directorate under the MoH which is taking the lead to guide the systems to be development and supported. The collaboration also benefits from the insight of experts, who successfully implemented an electronic data-collection tool in the tertiary care hospital in Blantyre (Surveillance Programme of IN-patients and Epidemiology –SPINE).[32] However, paper records are currently still an important part of recording and sharing health care data in Malawi, and the technical implementation of the DIPLOMATIC minimum datasets will take this into account. The DIPLOMATIC minimum datasets will ideally feed off structured patient records that include the variables of interest, but these may be paper based or electronic. Engagement with staff in the participating sites, to discuss advantages of different ways of data capture, to optimise ease of data recording, and to establish how this can be supported through existing clerking staff, will be essential to address the challenges of consistent data recording, in the face of pressures on clinical staff time.'

In the results section, the explanation on contemporaneous and retrospective data collection has been expanded as follows:

'To mitigate the risk of incomplete information due to patients receiving some elements of their care in facilities not participating in DIPLOMATIC, the datasets were designed for a hybrid approach between contemporaneous and retrospective data collection. Data would ideally be collected in each of the settings as care is completed (ANC and maternity and neonatal inpatient). However, the datasets also provide the option to capture retrospective summary information if the relevant data was not captured at the time the care was delivered. The maternity dataset thus includes a summary section on ANC, and the specialist neonatal care dataset allows retrospective recording of a summary of maternity/delivery care, if required (Figure 1). If the relevant data is captured electronically at the point of care, these retrospective summary sections can be auto-populated.'

The DIPLOMATIC minimum datasets have been designed so that the data collected can fully replicate the relevant government registers, in order to avert the risk of duplication of effort as highlighted by the reviewer. The MoH is planning to replace the manual register with eRegister and this will contribute to addressing this challenge. The Quality Management and Digital Health (QMDH) Directorate under Malawi's MoH is planning to use this eRegister platform to integrate all systems for primary health care. Currently the platform is underdevelopment and piloting is ongoing in 1 health facility in southern Malawi. This work is expected to be completed in July – August 2020. In recent discussions the Quality Management and Digital Health (QMDH) directorate under Malawi's MoH have provided their support to integrate the DIPLOMATIC minimal datasets into this eRegister platform. This has now been clarified at the end of the first paragraph of the discussion, which now states:

'Consultation and refinement will need to continue during pilots in the DIPLOMATIC partner facilities, planned to commence in autumn 2020, to ensure the datasets remain aligned with MoH reporting requirements and to enable synergies with projects in the fast moving field of electronic medical record (EMR) developments, such as an eRegisters platform currently being piloted. In recent discussions the Quality Management and Digital Health (QMDH) directorate under Malawi's MoH have provided their support to integrate the DIPLOMATIC minimal datasets into this eRegister platform.'

Putting "Routine" and standard of care into context of Malawi

As well described in the manuscript, the Malawi Ministry of Health has put many instruments in place to collect routine data. It should however be noted that routine data is frequently incomplete. This has been documented in other low-income countries ...

<https://journals.plos.org/plosone/article?id=10.1371/journal.pone.0211265>

<https://www.ncbi.nlm.nih.gov/pmc/articles/PMC4030005/>

Data incompleteness could have many possible causes, but one contributing factor is the degree to which the time required to capture data is sufficient for the person responsible. I interpret "routine" to mean that the work can be done without additional human resources. In the context of a Malawi government hospital (central or district) it is the clinical staff who record data such the variables shown in Tables 1-3. Malawi is know to have a crisis in human resources for health with the lowest number of doctors per capita and third lowest nurses per capital of any country (arguably old statistics but likely similar). Incompleteness of routine data is likely contributed to by overburdened healthcare workers. If the number of data elements that needs to be captures has increased it seems logical that data completeness could further decline. If I have misunderstood this and additional research staff will be hired to capture data in DIPLOMATIC partner sites, it would be good to mention this in the manuscript.

The statement in Table 3 that "many of these process measures are required for mandatory reporting to MoH" may be accurate, but conveys that "required" translates to "done", which I believe is not always the case. In ad hoc conversations with Malawian nurses and clinicians when asking about how a process is done the sentence frequently starts with "We are supposed to ...", followed by an explanation. When asking why they used the term "supposed to", the response is typically " You know due to pressures of work ..." or "Unfortunately the required materials are rarely available". There is a difference between desired standard of care and actual standard of care. These are of course shortcoming of the health system not shortcomings of the healthcare workers. I raise these points because I feel the sentiment of the term "routine" as used in the manuscript is intended to imply that it is a parts of "standard work" that healthcare providers are able to complete. However, I do not believe this to be the case in most government hospitals in Malawi.

As a final note on this point I believe it would be useful to look at the SPINE project that was undertaken at Queen Elizabeth Central Hospital described in this 2013 publication ...

<https://journals.plos.org/plosmedicine/article?id=10.1371/journal.pmed.1001400>

I believe lessons can be learned from this project.

We agree with the reviewer regarding the challenges posed by an extensive dataset. This is highlighted in the strengths and limitations sections (summary section, and within the discussion), which have been rephrased to identify the size of the dataset more clearly as a limitation:

In the summary section:

- *One limitation of this inclusive approach is that the datasets are more extensive than other exemplars of national minimum datasets.*
- *However the approach ensures that the datasets can integrate with existing systems and meet local data requirements, and also have the capacity to evaluate other Ministry of Health initiatives, rather than addressing research requirements only.*

Within the discussion:

'One limitation of the DIPLOMATIC datasets is that they are more extensive than other exemplars of national minimum datasets on maternal and neonatal health.[24, 25] This poses a challenge for implementation of these datasets in the context of stretched clinical staff, a background of predominantly paper based data recording and mixed success of electronic data systems.[10]

An additional paragraph has been added to the discussion, elaborating on plans for developing an appropriate system for data capture, recognising the challenges that this will pose. We agree with the reviewer that the lessons learned from SPINE are directly applicable to this work. The DIPLOMATIC collaboration includes experts who are involved in SPINE, and greatly benefits from their insights.

'DIPLOMATIC is staying abreast of the changing electronic health records landscape in Malawi, and is proactively engaging with the Quality Management and Digital Health Directorate under the MoH which is taking the lead to guide the systems to be development and supported. The collaboration also benefits from the insight of experts, who successfully implemented an electronic data-collection tool in the tertiary care hospital in Blantyre (Surveillance Programme of IN-patients and Epidemiology –SPINE).[32] However, paper records are currently still an important part of recording and sharing health care data in Malawi, and the technical implementation of the DIPLOMATIC minimum datasets will take this into account. The DIPLOMATIC minimum datasets will ideally feed off structured patient records that include the variables of interest, but these may be paper based or electronic. Engagement with staff in the participating sites, to discuss advantages of different ways of data capture, to optimise ease of data recording, and to establish how this can be supported through existing clerking staff, will be essential to address the challenges of consistent data recording, in the face of pressures on clinical staff time.'

We agree with the reviewer that reporting requirements for care processes by no means indicate that these can be delivered. As indicated in the results section, one of the refinements of the datasets arising from the consultation process was the inclusion of response levels indicating that the process could not be carried out. We thank the reviewer for identifying that a stronger emphasis on this point is required, and have now also highlighted this in the discussion section:

'The datasets include process measures on different elements of routine care. These elements serve to meet local reporting requirements, and by recording uptake of evidence based antenatal, maternal and neonatal care as recommended by WHO,[19-21] they facilitate monitoring of improvements in quality of care. However, each of these data elements also include response options to reflect that specific care in line with the protocols could not be carried out for example due to stock out of the relevant drugs, in recognition that resource limitations pose important barriers to implementation of care in line with these guidelines.'

Conveying inclusiveness of consultation and MoH involvement

While the manuscript makes frequent reference to involvement and consultation of the Ministry of Health, the Ministry has many Programmes within it, and these Programmes do not always operate vertically. Services such as diagnostics, pharmaceuticals, and immunization often cut across Programmes. Other Programmes intersect. For example the Reproductive Health Unit and the Department for HIV and AIDS both need access to information about PMTCT. Historically

Programmes have struggled to align their intersecting data needs. It suggest adding some mention in the methods section to understand how this unified minimum data set has addressed the needs of the Expanded Program for Immunization, the Reproductive Health Unit and the Department of HIV and AIDS.

The methods section specifies the relevant units and departments within the Ministry of Health, with whom consultation took place. We concur that several elements of the data to be collected are relevant for other cross cutting programmes, such as the Expanded Programme for Immunization and the Department for HIV and AIDS. However, the relevant variables were adopted into the DIPLOMATIC minimum datasets unchanged from current Ministry of Health specifications for registers and monthly returns for antenatal, maternity and neonatal care, and thus no specific consultation with these programmes/ departments was carried out. We recognise that ongoing close working with the Ministry of Health will be vital for the success of the collaboration, and are closely following the advice of the Reproductive Health Directorate and the Quality Management and Digital Health Directorate on the appropriate engagement.

General comments about patient identification

There is no mention of the national registration system that was recently introduced in Malawi providing Identification cards to everyone over 16yo. While there are still gaps in coverage, and some mothers are younger than 16, the National ID would seem like an obvious mechanism to uniquely identify mothers. Furthermore the QR code that is printed on the back of the national ID card lends itself to fast and reliable electronic capture of demographic details.

We have now elaborated on this in the results section of the manuscript as follows:

'Malawi is in the process of implementing a unique health identification (UHID) system. Generation of a UHID will be facilitated through registration at the point of care, based on verification of identity using the recently introduced national ID (or birth certification). Use of a Quick Response (QR) code for ease and speed of identification of the patient is being considered by the MoH. Each of the three DIPLOMATIC datasets allows recording of the local health record identifier, and this will be used to record the UHID as it embeds into health care record use in Malawi. Consistent use of the UHID across records of care will greatly support data linkage and may eventually allow automated imports of demographic details into the patient record, thus reducing data entry effort. However experience from Scotland shows that linkage of maternal and neonatal health records can remain challenging, even when a unique health identifier is well established. The demographic and ID section of each of the three DIPLOMATIC datasets, is designed to facilitate data linkage, even in the absence of or in the event of a failure of linkage through a UHID.'

Adding details to the results section

The results do not convey the magnitude of the change in the number of variables that need to be routinely collected. This makes it difficult to assess the change in level of effort required to record the information. For example did the number of variables go up by 20%, 100%, 200%? I think one or more graphics (flow diagram) could be a nice addition. Datasets were taken from multiple sources. For each of the three minimum datasets created, show how many variables were in the initial MoH routinely collected data, how many variables were in the WHO recommended dataset, and how many in the intersection of the two datasets. For each subsequent step of the process (comparison with Scottish MDS/consultations/expert reviews/ ???) show how this impacted the number of data elements. A flow diagram along the lines of this graphic https://www.researchgate.net/figure/Flow-chart-outlining-selection-of-papers-forreview_fig1_256836094.

On page 9, line 38 the manuscript mentions "refine" the dataset. Specifically what does that mean in terms of additions/deletions/substitutions?

We have created a flow diagram on the development of the datasets (now Figure 1), as suggested.

Strengths and limitations

There is a strengths and limitations section on Page 6 as per the BMJ Open format. These all appear to be strengths, with no apparent limitations listed. Suggest moving bullet #4 to the end and editing to reflect a possible limitation ... “[one limitation is that] the datasets are more extensive than other exemplars of national minimum datasets [and therefore will be more burdensome to collect. [However] inclusion of additional variables ensures they can integrate with existing systems and meet local data requirements, rather than addressing research requirements only, and also have the capacity to evaluate other Ministry of Health initiatives.”

The bullet point was moved to the end and rephrased in line with the reviewers comments, which we agree helps to clarify that this is a limitation:

- *One limitation of this inclusive approach is that the datasets are more extensive than other exemplars of national minimum datasets.*
- *However the approach ensures that the datasets can integrate with existing systems and meet local data requirements, and also have the capacity to evaluate other Ministry of Health initiatives, rather than addressing research requirements only.*

The strengths and limitations on Page 20 the term “systematic review” is used. I believe a formal systematic review has a protocol (e.g search terms used, etc.) and this was not described in the methods section. If a formal systematic review was indeed done then more details need to be added in the methods section. If it was not done then using an alternative term would be appropriate.

We thank the reviewer for highlighting the inconsistent use of this phrase. This section has now been brought in line with the description of the process as given in the introduction, and now reads:

‘In line with recommended practice, the datasets were designed using a multimodal approach, combining an in depth review of the contextual information and published as well as grey literature with expert input.’

Graphics

I am unclear on the meaning of the color-coding (black ,red, green) in Figure 1. The shadowing on the text and the poor contrast between the colors make is somewhat difficult to read. I am unclear why some boxes are dark green and other boxes light green. The purpose of the lines that span the three columns is unclear to me.

The different colours merely differentiated the overall title (black), the titles of the three datasets (red) and the components of the datasets (green), (choice of colours picked up on the colours of the Malawian flag).

The lines, connecting the ID and demography components of the three datasets, indicate that these facilitate the linkage between the datasets. The shades of green differentiate the summary sections for previous care from the prospective data elements, with the lines spanning the columns indicating the source of the corresponding elements for the summary, if this was recorded at the time.

We have elaborated on the figure description in the heading and legend. A revised figure in black and white, with improved contrast is being submitted with the revised document (now Figure 2).

Notes regarding the tables

Table 1

What is a “Booking contact”? (also Table 2)

First (booking) visit as captured in the ANC Register, at which a full history is taken and, initial screening for medical, psychological and social risk factors takes places. Estimated date of delivery will also be recorded at the booking visit, although this may subsequently be amended if the first ultrasound scan for the pregnancy takes place after the booking visit.

This explanation has been added as a footnote to Table 1 and 2.

Table 2

In the footnote “*Karonga Health and Demographic Health Surveillance System (HDSS)[27]” and asterisk is used where as in the body of the table is says “HDSS 1” (also Table 3)

This has been corrected to be an asterisk within Table 2 and 3, linking to the corresponding footnote.

Table 3

“final EDD” presumably refers to actual delivery date now estimated.

EDD should be recorded at the booking contact and should only be updated once, and only if the first ultrasound scan takes place after the booking visit. Final EDD in Table 3 relates to the last (final) estimated date of delivery, as determined as part of ANC. Gestation at birth is based on the difference between the estimated and actual date of delivery.

In the ANC record and the maternity discharge record, more details about EDD are recorded (method of estimation, and whether there was one change to the EDD, as described above). For the specialist neonatal dataset, only the final EDD without these details are recorded. This explanation exceeds the level of detail required for the high level description of the data elements, so the wording in table three has been changed to just ‘EDD’, to avoid confusion.

References

Ref #25, Price, missing initial “A”.

Ref #26, et al.. (two full stops).

The reference list was corrected accordingly.

VERSION 2 – REVIEW

REVIEWER	Deepa Dongarwar Baylor College of Medicine, USA
REVIEW RETURNED	28-Jul-2020

GENERAL COMMENTS	The authors have sufficiently addressed all the previous queries and concerns. Thanks for taking the time and efforts to do so.
---

REVIEWER	Gerald Douglas University of Pittsburgh, USA.
REVIEW RETURNED	08-Aug-2020

GENERAL COMMENTS	I believe you have address nearly all my comments from the first review. line 53 suggest "low-income country setting". low-income or low- and middle-income typically hyphenated like this. I think Figure 1 provides a clear overview of the process and timeline. Well done. The strengths and limitation section reads like a paragraph with each sentence turned into a bullet, rather than a set of independent statements. Starting a bullet with "instead", or however" suggests the content is actually part of the prior bullet. Suggest trying to reword.
---